# The cost-effectiveness of controlling cervical cancer using a new 9-valent human papillomavirus vaccine among school-aged girls in Australia

Rashidul Alam Mahumud[1,2,3,4]*, Khorshed Alam[1,2], Jeff Dunn[1,5,6], Jeff Gow[1,2,7]

**1** Health Economics and Policy Research, Centre for Health, Informatics and Economic Research, University of Southern Queensland, Toowoomba, Queensland, Australia, **2** School of Commerce, University of Southern Queensland, Toowoomba, QLD Australia, **3** Health Economics Research, Health Systems and Population Studies Division, International Centre for Diarrhoeal Disease Research, Bangladesh (icddr,b), Dhaka, Bangladesh, **4** Health and Epidemiology Research, Department of Statistics, University of Rajshahi, Rajshahi, Bangladesh, **5** Cancer Research Centre, Cancer Council Queensland, Fortitude Valley, QLD, Australia, **6** Prostate Cancer Foundation of Australia, St Leonards NSW, Australia, **7** School of Accounting, Economics and Finance, University of KwaZulu-Natal, Durban, South Africa

* rashed.mahumud@usq.edu.au, rashidul.icddrb@gmail.com

**Data Availability Statement:** All relevant data are within the manuscript and its Supporting Information files.

## Abstract

### Introduction

Cervical cancer imposes a substantial health burden worldwide including in Australia and is caused by persistent infection with one of 13 sexually transmitted high-risk human papillomavirus (*HPV*) types. The objective of this study was to assess the cost-effectiveness of adding a nonavalent new Gardasil-9® (*9vHPV*) vaccine to the national immunisation schedule in Australia across three different delivery strategies.

### Materials and methods

The Papillomavirus Rapid Interface for Modelling and Economics (*PRIME*) model was used to examine the cost-effectiveness of *9vHPV* vaccine introduction to prevent *HPV* infection. Academic literature and anecdotal evidence were included on the demographic variables, cervical cancer incidence and mortality, treatment costs, and vaccine delivery costs. The incremental cost-effectiveness ratios (*ICERs*) were measured per disability-adjusted life years (*DALYs*) averted, using the heuristic cost-effectiveness threshold defined by the World Health Organisation (*WHO*). Analyses and data from international agencies were used in scenario analysis from the health system and societal perspectives.

### Results

The 9vHPV vaccination was estimated to prevent 113 new cases of cervical cancer (discounted) during a 20-year period. From the health system and societal perspectives, the 9vHPV vaccination was very cost-effective in comparison with the status quo, with an ICER of A$47,008 and A$44,678 per *DALY* averted, respectively, using the heuristic cost-

**Funding:** The authors received no specific funding for this work.

**Competing interests:** The authors have declared that no competing interests exist.

effectiveness threshold level. Considering delivery strategies, the *ICERs* per *DALY* averted were A$47,605, A$46,682, and A$46,738 for school, health facilities, and outreach-based vaccination programs from the health system perspective, wherein, from the societal perspective, the *ICERs* per *DALY* averted were A$46,378, A$43,729, A$43,930, respectively. All estimates of *ICERs* fell below the threshold level (A$73,267).

## Conclusions

This cost-effectiveness evaluation suggests that the routine two-dose *9vHPV* vaccination strategy of preadolescent girls against *HPV* is very cost-effective in Australia from both the health system and societal perspectives. If equally priced, the *9vHPV* option is the most economically viable vaccine. Overall, this analysis seeks to contribute to an evidence-based recommendation about the new *9vHPV* vaccination in the national immunisation program in Australia.

## Introduction

Cervical cancer is both a leading cancer and the leading cause of cancer deaths in women globally [1]. An estimated 570,000 new cases of cervical cancer were diagnosed in 2018, composing 6.6% of all cancers in women [1]. In Australia, over the last couple of years, the age-specific cervical cancer incidence has slightly reduced to 7.1 cases per 100,000 females in 2018 from 7.4 cases per 100,000 females in 2014 [2]. However, the incidence is quite high among young adult females at 15.0 cases per 100,000 females and was the most frequently diagnosed cancer among women in 2018 [2]. Persistent infections with human papillomavirus (*HPV*) are a key cause of cervical cancer and an established carcinogen of cervical cancer [3]. *HPV* is predominantly transmitted to reproductive-aged women through sexual contact [4]. Most HPV infections are transient and can be cleared up within a short duration, usually a few months after their acquisition. However, *HPV* infections can continue and evolve in cancer in some cases. There are more than 100 types of *HPV* infections that have been identified and divided into low- and high-risk types develop into cervical cancer [5]. Thirteen high-risk HPV types are known to be predominantly responsible for malignant and premalignant lesions of the anogenital area [6] and are the leading causes of most aggressive cervical cancers [7]. Further, *HPV* is also responsible for the majority of anogenital cervical cancers such as anal cancers (88%), vulvar cancers (43%), invasive vaginal carcinomas (70%), and all penile cancers (50%) globally [5]. The incidence of neck and head cancers caused by *HPV* infection is low but not negligible [8]. Cervical cancer is preventable through implementation of a primary prevention strategy such as vaccination worldwide including Australia [9,10]. Therefore, a reduction in cervical cancer incidence and associated cancer mortality along with the improvement of survival rates have the potential to reduce the burden of cervical cancer.

The high burden of cervical cancer in terms of incidence and associated mortality rates across the world could be reduced by incorporating a comprehensive primary prevention mechanism. Prevention mechanisms includes early vaccination, diagnosis, effective screening, adequate referral and advanced course of treatment procedures. In this context, *HPV* vaccinations (i.e., *bivalent and quadrivalent*) has been introduced in many countries in the past decade [10]. Currently, available *HPV* vaccines can promote herd immunity against cancer-causing types of *HPV* that helps to reduce the high-risk of cervical cancer burden. These vaccines have

played a significant role in preventing *HPV* infection types 16 and 18 [10], which cause more than 70% of cervical cancers in Australia [7].

Australia was the first country to implement a publicly-funded National *HPV* Immunisation Program (*NHIP*), starting with preadolescent girls in 2007, using the quadrivalent Gardasil® vaccine (*4vHPV*; Merck & Co., Kenilworth, NJ, USA) [11]. The goals of the immunisation program were to reduce the acquisition and spread of *HPV* infections and to achieve optimum coverage through the school-based delivery system [12]. This program for adolescent employed a three-dose schedule of the 4vHPV vaccine [13]. The *4vHPV* vaccine provides protection against *HPV* infection types 6, 11, 16, and 18 [14]. In the context of Australia, the *4vHPV* vaccine was replaced by the two-dose nonavalent Gardasil®-9 vaccine (*9vHPV*; Merck Sharp & Dohme) in 2018 [15]. According to the underlying distribution of *HPV* infection types of cervical cancers, the *9vHPV* vaccine builds population-level strong immunity against *HPV*-6, 11, 16, 18, 31, 33, 45, 52, and 58 infections [6] that cumulatively contribute to approximately 89% of all cervical cancers globally [16] and 93% in Australia [17]. Considering the primary prevention of *HPV* infection, the *9vHPV* vaccine is anticipated to reduce by 10% more the lifetime risk of diagnosis of cervical cancer in immunised cohorts than the *4vHPV* vaccine and by 52% more compared to non-vaccinated cohorts [18].

With the availability of vaccines against the different *HPV* infection types, there are good opportunities for primary prevention to add to continuing efforts on secondary prevention strategies. However, the decision for any country to add a new vaccine to national immunization programs requires careful assessment of the relative value of the vaccine compared with alternative uses of the required resources (i.e., cost-effectiveness) and its affordability (i.e., budgetary impact). Cost-effectiveness analysis (*CEA*) is a pragmatic approach which aims to examine the outcomes and costs of interventions or programs designed to improve health. *CEA* evolves measuring the net or incremental costs and effects of an intervention or program in terms of costs and health outcomes as compared with some comparator. There is considerable evidence of assessing the cost-effectiveness of the *9vHPV* vaccine in different country settings. In Canada, the *9vHPV* was found to be highly cost-effective compared with the *4vHPV* vaccine taking into consideration the shorter duration of protection (*9vHPV* = 20 years vs. *4vHPV* = lifelong), along with a lower vaccine efficacy (85% vs. 95%) [19]. In other studies conducted in the United States (*US*), the *9vHPV* vaccine was also found to be very cost-effective compared to the *4vHPV* vaccine [20]. However, findings from cost-effective evaluations will differ based on study settings, funding, perspectives and coverage of vaccination For example, in the *US*, Chesson et al. (2016) found that the *9vHPV* vaccine was not cost-effective, with an incremental cost-effectiveness ratio (*ICER*) of $146,200 per quality-adjusted life year (*QALY*) gained that exceeded the cost-effectiveness threshold ($100,000) [21]. Some cost-effective evaluations were performed using the same vaccine (i.e., 9-valent) in the *US* to capture the different dimensions of its economic viability [22–27]. These studies incorporated different study participants, designs, perspectives, vaccine delivery routes and model specifications. Simms et al. (2016) evaluated the *9vHPV* vaccine in a primary *HPV* screening scenario in both Australia and Canada [18]. They found that *9vHPV* had a significant impact on reducing cervical cancer incidence from the health system perspective. Further, they claimed that the incremental cost per dose in girls should not exceed a median of A$35.99. However, this study emphasised the impact of vaccines to prevent cervical cancer rather than their economic viability. Sufficient evidence did not arise for health policymakers to use the findings to develop cost-effective intervention strategies. In Germany, universal immunisation with *9vHPV* was suggested as it had an *ICER* of €22,987/*QALY* gained, which was below the threshold [28]. In Spain, a recent study evaluated a vaccine program in adolescent girls, wherein the *9vHPV* vaccine was found to be more highly cost-effective, with an *ICER* of €7,718 per *QALY* compared to the *4vHPV*

vaccine [29]. In the African setting, in Kenya and Uganda, a study recommended that the *9vHPV* vaccine was very cost-effective in both countries, wherein the additional cost of the *9vHPV* vaccine did not exceed I\$8.3 per immunised girl [30].

In Australia, the *9vHPV* vaccine was introduced in 2018. There is limited current comprehensive evidence about the cost-effectiveness of the *9vHPV* vaccine in Australia across delivery strategies (e.g., school-based, health facility-based and outreach-based) from the health system and societal perspectives. The present study evaluates the cost-effectiveness of the *9vHPV* vaccine from both health system and societal perspectives across three delivery routes. However, the previous cost-effective evaluation considered only one perspective nor health system or societal, or both perspectives along with single vaccine delivery route. Further, the findings of the present study will provide evidence about the cost-effectiveness of the *9vHPV* vaccine to policymakers. These cost-effectiveness findings will also be significant for determining the optimal pricing of delivery strategies in the vaccination program in order to maximise the societal benefits of the introduction of the new *9vHPV* vaccine to Australia.

The objectives of this study are (1) to examine the cost-effectiveness of the *9vHPV* vaccine by considering three different vaccine delivery strategies in the setting of Australia from the health system and societal perspectives and (2) compare the *ICER* per case, disability-adjusted life years (*DALYs*), and life-years saved across delivery strategies such as school-based, health facility-based, and outreach-based programs.

## Materials and methods

### Study perspective

This study was designed from both the health system and societal perspectives. The societal perspective refers to all types of costs that can be identified, quantified, estimated, and valued no matter who incurred them and it is considered to be the summation of both provider and household costs. This is the recommended standard for undertaking cost-effectiveness analysis [31].

### Model overview

The study used the Papillomavirus Rapid Interface for Modelling and Economics (*PRIME*) model. *PRIME* is a user-friendly model designed and developed by the World Health Organisation (*WHO*) in collaboration with the Johns Hopkins Bloomberg School of Public Health in Baltimore, the School of Hygiene and Tropical Medicine in London, and the Universite Laval in Quebec [10]. *PRIME* is a Microsoft Excel based (Microsoft Corp., Armonk, NY, USA) static model that measures the health and economic effects of the vaccination of adolescent girls against *HPV* infection. It is not designed to examine other dimensions, such as immunised males or older women or the impact of cervical cancer screening services [10]. Several spreadsheets are contained in this model to input different parameter-level data on demographics, an age-dependent incidence of cervical cancer, associated mortality, vaccine efficacy, vaccine coverage, and associated costs (e.g., vaccination costs, treatment costs). This model does not consider indirect effects like herd immunity.

### Methodological assumptions

Methodological assumptions follow the *WHO* guidelines for cost-effectiveness analysis [32]. The use of cost-effectiveness analysis is recommended when considering health system and societal perspectives. In the context of the health system perspective, the average cost parameters associated with treating a woman with cervical cancer (per episode, over the lifetime), and the cost of the *HPV* vaccination program were both considered. From the societal viewpoint,

both direct medical (e.g., drugs, diagnostics) and non-medical (e.g., transportation) costs as well as indirect costs (e.g., productivity loses or income loss due to cervical cancer) were considered in the analysis. All future costs and health benefits were adjusted by a discount rate of 5% annually [5,29,32], which was validated in the sensitivity analysis. The primary outcome measure is the *ICERs* per *DALYs* averted. *DALY* estimation was undertaken by summing up the fatal burden (years of life lost; *YLL*) due to premature cervical cancer related mortality and the non-fatal burden (years lost due to disability; *YLD*) for patients surviving the condition.

$$DALY = YLL + YLD \tag{1}$$

$$YLL = \frac{N}{r} \left(1 - e^{-rL}\right) \tag{2}$$

$$YLD = I \times DW \times L \left(\frac{1 - e^{-rL}}{r}\right) \tag{3}$$

where, N = number of deaths; L (*YLL*) = standard life expectancy at the age of death in that year; I = number of people with cervical cancer cases; *DW* = disability weight; r = discount rate; and L (*YLD*) = duration of disability in years.

## Cost-effectiveness analysis

The performance of competing strategies was explained using the *ICER* which were calculated by dividing the difference in cost with and without HPV vaccination by the difference in health outcomes (e.g., the number of *DALYs* averted, the number of deaths and cases averted) with and without vaccination in Australia. The *ICER* is used to examine whether the *9vHPV* vaccine is economically viable in Australia. In the context of Australia, no explicit cost-effectiveness threshold has been approved [33,34], although research has confirmed that there is a correlation between the incremental cost per health outcomes (e.g., *QALY* gained or *DALY* averted) and the probability of rejection of a health intervention or a new medicine [35]. The pharmaceutical industry claim that an acceptable threshold was in the range of *AUD* 45,000 to *AUD* 60,000 per additional *QALY* gained [36]. Some studies also stated that "Pharmaceutical Benefits Advisory Committee (*PBAC*) decisions in the past have shown that the *ICER* per *QALY* gained was of the order of $50,000" [18,37]. The present study intended to evaluate the cost-effectiveness of the *9vHPV* vaccine in terms of the *ICER* per *DALYs* averted. Further, *DALYs* and *QALYs* differ in concept and application. The concept of *DALYs* was used to measure the disease burden using life lost due to premature death and the time spent in worse healthy states. Empirical evidence in the Australian context is limited to the use of the willingness-to-pay (*WTP*) threshold values for the *ICER* per *DALYs* averted. In reporting the cost-effectiveness scenario, the present study used the heuristic cost-effectiveness threshold as defined by the *WHO* Commission on Macroeconomics and Health (*CMH*) [38]. The gross domestic product (*GDP*)-related cost-effectiveness thresholds were based on assumptions about leisure time, non-health consumption, longevity and health-related quality of life. An intervention is cost-effective if the *ICER* per *DALY* averted is less than three times of a country's annual per capita GDP. According to this guideline, the *CMH* recommended three broad decision rules, as follows: (1) a program or intervention is defined as very cost-effective if the *ICER* per *DALY* averted is less than one time the *GDP* per capita; (2) a program or intervention is cost-effective if the *ICER* per *DALY* averted is one or more times the *GDP* per capita but less than or equal to three times the *GDP* per capita; and (3) a program or intervention is not cost-effective if the *ICER* per *DALY* averted is more than three times the *GDP* per capita [31].

## Vaccine and efficacy

The Australian Technical Advisory Group on Immunisation (*ATAGI*) in 2018 advised moving from using the quadrivalent *4vHPV* to using the nonavalent Gardasil-9 (*9vHPV*) vaccine [39]. The *9vHPV* vaccine has been registered for use in Australia [40]. The vaccine is funded through the national immunisation program (*NIP*) and delivered primarily by state and territory school-based immunisation programs in Australia [39]. This vaccine is manufactured using a procedure similar to that of the *4vHPV* vaccine, which contains 0.5mg of aluminium hydroxyphosphate sulphate and a yeast expression system [40]. The *4vHPV* vaccine contains five more virus-like particles than the original vaccine, identical to those in the protective capsule around the nine included strains (*HPV*-6, 11, 16, 18, 31, 33, 45, 52, and 58) with the aim to further reduce the *HPV* disease burden. The high prophylactic efficacy of the *9vHPV* vaccine (93%) against *HPV* infection is evident both in Australia (77% for *HPV* types 16, 18 and 16% for *HPV* types 6, 31, 33, 45, 52, 58) [17] and globally (89%) [16]. However, no herd immunity was considered. It was recommended for the target cohort of adolescents aged 12 years to receive a two-dose *9vHPV* vaccination for several reasons [39]. First, administering a vaccination at this age is more likely to ensure it is being given before their first sexual encounter (and *HPV* exposure). Also, the immune response tends to be stronger and more long-lasting when the vaccine is given to pre-adolescents. However, *9vHPV* is not recommended for use during pregnancy. Similarly, vaccination is delayed if the person is unwell or has a high temperature, medical advice is recommended if the person is allergic to yeast or has had a severe reaction to a previous vaccine, and anyone who receives the vaccine is recommended to sit for 15 minutes thereafter to reduce the risk of fainting.

## Vaccine delivery strategies

The *HPV* vaccine delivery strategy is an important aspect that needs to be considered carefully by each country. According to the country-specific context, the costs of vaccine delivery may vary. The *WHO* has recommended several types of common vaccine delivery strategies for different country settings. One example is vaccine delivery at healthcare facilities and via outreach routes (e.g. school-based program) and campaigns. It may be required to use a combined vaccine delivery strategy to ensure access among the entire target population. The *9vHPV* vaccine has been delivered in Australia through school-based *NIP* in all states and territories to the target population cohort of school-going adolescents since January 2018. Two doses of *9vHPV* are recommended to be administered at a minimum interval of six to 12 months between doses [39]. In some cases, general practitioner (*GP*) and other primary health care providers are generally engaged to catch up doses missed in the routinely school-based *NIP*. All providers are proactively involved in delivering and ensuring the completion of all doses of the *9vHPV* vaccine to those individuals with special requirements, vaccine hesitancy, or immunocompromise. However, individuals who have already been fully immunised with HPV vaccines are not eligible for free *9vHPV* vaccination. The present study incorporated another two hypothetical vaccine delivery strategies, as health facility-based and outreach-based, both from the health system and societal perspectives.

## Vaccine delivery costs

The present study considered vaccine delivery related costs across three delivery strategies (e.g., school-based, health facility-based and outreach-based). Costs were derived from an existing costing study [41]. This study captured both financial and economic costs according to the *WHO* guidelines [42], included eight cost parameters, and focused on the investment and recurrent cost impacts of *HPV* vaccination on existing vaccination services. Furthermore,

the investment costs were defined as microplanning (e.g., per diems and travel allowances, venue rental, transport and personnel time spent), training (e.g., training materials, stationery), social mobilisation (e.g., facilitator time in meetings, production of television/radio spots, posters, leaflets, value of teacher and volunteer time), and cold chain supplement. In addition, recurrent costs were covered including vaccines, service delivery, monitoring and evaluation, and waste disposal.

### Cervical cancer treatment costs

**Direct medical costs.** Cervical cancer treatment costs were derived from a previous cost-of-illness study considering four treatment procedures: localised cancer treatment, regional cancer treatment, distant cancer treatment and terminal care [43]. The treatment costs were estimated based on different parameters such as surgical (e.g., conisation, hysterectomy, radical hysterectomy) and non-surgical (e.g., radiation therapy, adjuvant radiation therapy, chemo-radiation) [43]. Different types of activities included in cancer diagnosis were the direct medical costs such as colposcopy, chest X-ray, computerised tomography scan, positron-emission tomography scan, magnetic resonance imaging, bone scan and cystoscopy. Other costs included those related to inpatient care, emergency care, medicine costs, rehab, complex continuing care, long-term care, home care services, physician consultations, and non-physician provider costs.

**Indirect costs.** Indirect costs of cervical cancer patients and vaccine receivers were restricted to the loss of labour productivity due to ill health. Absenteeism-related data were obtained for a cervical cancer episode from elsewhere [32]. Other indirect costs were estimated using the human capital approach (S1 Table). The production losses were measured in both monetary and quantitative terms (e.g., days of productivity loss) [44]. The value of unpaid time devoted to own care and family defined caregivers [45]. The value of daily productivity was measured based on an age-specific average wage [46]. The average daily wage of cervical cancer patients were used for adult patients, and one-half of that wage was applied to teenager patients. Intangible costs related to pain, discomfort and grief were excluded [46]. All costs were converted into 2018 Australian dollars using the Consumer Price Index of Health Care [47].

**Dynamic modelling of HPV transmission and the impact of vaccination.** A dynamic cancer disease model was introduced to cover *HPV* transmission, *HPV* vaccination and cervical pre-cancer (Fig 1). The model incorporates demographics, economics, *HPV* attributable fractions in cervical cancer and vaccine uptake assumptions, as detailed in Table 1. When modelling the impact of *HPV* vaccination, the model captured the effects of herd protection (i.e., naturally acquired immunity) on the unvaccinated cohort. It was assumed that *9vHPV* vaccine type-specific (*HPV* types) efficacy in girls was 100% and that the duration of protection was 20 years [19,27].

**Sensitivity analysis.** A deterministic sensitivity analysis was performed to examine the robustness of the results. The output estimates varied for each value of the input parameters. These prices were derived from the academic and anecdotal literature and aimed to determine the impact of uncertainty in input assumptions on the *ICERs*.

## Results

### Model input parameters

Table 1 shows several input parameters, including the population cohort at birth, coverage of full dose vaccine, vaccine effectiveness versus *HPV*-9 types, the price of vaccine, and vaccine delivery costs per fully immunised girl. Cervical cancer treatment related costs per episode

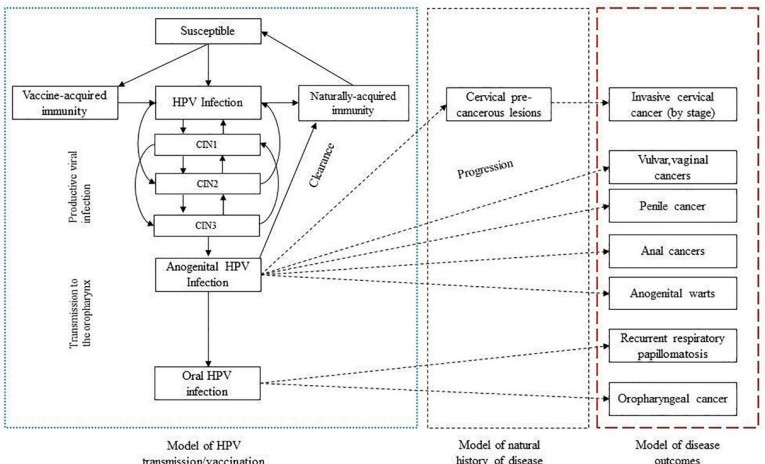

**Fig 1. Simplified diagram of the model of HPV transmission, human impact of vaccination and disease outcomes in Australia.**

included direct costs (e.g., medical and non-medical costs) and indirect costs (e.g., loss of labour productivity for patients and caregivers during treatment). *DALYs* incurred for nonfatal and fatal cervical cancer episodes, and epidemiological data related to cervical cancer incidence were used. The sizes of the female birth cohort and the cohort at immunisation age were 191,340 and 118,679, respectively. Vaccination coverage was 86%, whereas vaccine effectiveness against *HPV* infections was 95%. The price of the vaccine and direct and indirect vaccine delivery costs were A\$280, A\$31.77, and A\$17.59, respectively. Cervical cancer treatment costs were A\$61,272, wherein 53.78% (A\$32,952) were direct costs and 46.22% (A\$28,322) indirect costs. These varied depending on the types of treatment and stages of cancer. Cervical cancer incidence and mortality-related data were extracted from national sources and the *GLOBO-CAN*-2018 study [48]. Methodological assumptions such as disability weights (for cancer diagnosis, non-terminal and terminal) are shown in Table 1. Vaccine protection was considered to be 20 years as suggested by an expert panel and earlier research [49].

## Cost-effectiveness estimates

The model estimates in Table 2 show the cost-effectiveness of the *9vHPV* vaccination in Australia under various assumptions about the cost of cervical cancer treatment and the cost of vaccination across delivery strategies. The *9vHPV* vaccination in Australia would cost the public approximately A\$28.11 million for this target population cohort, although several types of treatment procedures would be transferred from the health system perspective and the value of A\$26.72 million from a societal perspective across the various vaccine delivery strategies (e.g., school-based, health facility-based, outreach-based) would result. State and territory school-based immunisation programs primarily implement the *9vHPV* vaccination through the *NIP* in Australia. Another two possible delivery strategies (e.g., health facility-based and outreach-based) were also included for comparison. Overall, the *ICER* per *DALY* averted was A\$47,008 from a health system perspective and A\$44,678 from a societal perspective, respectively. Considering delivery strategies, the *ICERs* per *DALY* averted were A\$47,605, \$46,682 and \$46,738 for school-based, health facility-based and outreach-based programs, respectively, from the health system perspective. Whereas, from the societal perspective, the values were A\$46,378, A\$43,729, and A\$43,930 respectively. Both perspectives for *ICERs* per *DALY* averted fell below the 2018 fiscal year *GDP* per capita in Australia (A\$73,267), which is used as a

**Table 1.  Input parameter assumptions and sensitivity analysis.**

| Input parameters | Health system perspective | | | | Societal perspective | | | | Sensitivity analysis and potential sources |
|---|---|---|---|---|---|---|---|---|---|
| | Overall | School-based | Health facilities-based | Outreach-based | Overall | School-based | Health facilities-based | Outreach-based | |
| **Population** | | | | | | | | | |
| Population cohort at birth (female) ('000) | 191.34 | 191.34 | 191.34 | 191.34 | 191.34 | 191.34 | 191.34 | 191.34 | [50] |
| Population cohort at vaccination age (female) ('000) | 118.68 | 118.68 | 118.68 | 118.68 | 118.68 | 118.68 | 118.68 | 118.68 | [40,51] |
| Target age group (yrs) | 12 | 12 | 12 | 12 | 12 | 12 | 12 | 12 | [39] |
| **Vaccination and vaccine delivery costs** | | | | | | | | | |
| Vaccination coverage (full doses) | 86% | 86% | 86% | 86% | 86% | 86% | 86% | 86% | 72.00% -90.1% [51–55] |
| Vaccine effectiveness vs HPV types[1] | 95% | 95% | 95% | 95% | 95% | 95% | 95% | 95% | 85% -100% [25,52,53,56] |
| Price of vaccine per fully immunised girl (*FIG*) (A$) | 280 | 280 | 280 | 280 | 280 | 280 | 280 | 280 | 270–320 [32,41,57] |
| Direct costs of vaccine delivery per *FIG* (A$) | 31.77 | 35.27 | 29.86 | 30.19 | 31.77 | 35.27 | 29.86 | 30.19 | [5,32,41] |
| Indirect costs of vaccine delivery per *FIG* (A$) | - | - | - | - | 17.59 | 24.05 | 13.93 | 14.79 | |
| Total cost of vaccine delivery cost per *FIG* (A$) | 31.77 | 35.27 | 29.86 | 30.19 | 49.36 | 59.32 | 43.80 | 44.98 | |
| Total costs of vaccination per *FIG* (A$) | 311.77 | 315.27 | 309.86 | 310.19 | 329.36 | 339.32 | 323.80 | 324.98 | 300–500 [5,32,41] |
| **Treatment cost per episode** | | | | | | | | | |
| Direct costs A$ ('000) | 32.95 | 32.95 | 32.95 | 32.95 | 32.95 | 32.95 | 32.95 | 32.95 | [32,43] |
| Indirect costs (including caregiver costs) A$ ('000) | - | - | - | - | 28.32 | 28.32 | 28.32 | 28.32 | [32,43] |
| Total treatment costs A$ ('000) | 32.95 | 32.95 | 32.95 | 32.95 | 61.27 | 61.27 | 61.27 | 61.27 | 36.05–71.05 [32,43] |
| **Methodological assumptions** | | | | | | | | | |
| Disability weight for cancer diagnosis | 0.09 | 0.09 | 0.09 | 0.09 | 0.07 | 0.09 | 0.09 | 0.09 | 0.061–0.095 [10,32,58,59] |
| Disability weight for non-terminal (per year) | 0.08 | 0.08 | 0.08 | 0.08 | 0.08 | 0.08 | 0.08 | 0.08 | 0.065–0.091 [10,32] |
| Disability weight for terminal cancer | 0.8 | 0.8 | 0.8 | 0.8 | 0.8 | 0.8 | 0.8 | 0.8 | 0.70–0.90 (assumption) |
| Vaccine protection (years) | 20 yrs | 20 yrs | 20 yrs | 20 yrs | 20 yrs | 20 yrs | 20 yrs | 20 yrs | 20 years [19] |
| Discount rate | 5.0% | 5.0% | 5.0% | 5.0% | 5.0% | 5.0% | 5.0% | 5.0% | 3.0% -5.0% [23,24,30,32,58,60] |
| Proportion of cervical cancer cases that are due to [1]HPV-types | 90.3% | 90.3% | 90.3% | 90.3% | 90.3% | 90.3% | 90.3% | 90.3% | 70% -95.0% [9,15,17,39] |
| **Economic growth** | | | | | | | | | |
| GDP per capita, A$ | 73,267 | 73,267 | 73,267 | 73,267 | 73,267 | 73,267 | 73,267 | 73,267 | 73,267 [61] |

[1]HPV-6, 11, 16,18,31,33,45,52,58

threshold for examining the cost-effectiveness of an intervention. Similarly, consistent results were presented for the *ICERs* per life-year saved for both perspectives across delivery strategies (Table 2). This evaluation signifies the cost-effectiveness of the *9vHPV* vaccination from both perspectives in Australia.

**Table 2. Outcomes of the vaccination program\*.**

| Scenario | Scenario– 1 | | | | Scenario—2 | | | |
|---|---|---|---|---|---|---|---|---|
| Perspective | Health system perspective | | | | Societal perspective | | | |
| Vaccine delivery strategies | Overall | School-based | Health facilities-based | Outreach-based | Overall | School-based | Health facilities-based | Outreach-based |
| Output parameters | | | | | | | | |
| Cohort size at birth (female), ('000) | 191.34 | 191.34 | 191.34 | 191.34 | 191.34 | 191.34 | 191.34 | 191.34 |
| Cohort size at vaccination age (female) ('000) | 118.68 | 118.68 | 118.68 | 118.68 | 118.68 | 118.68 | 118.68 | 118.68 |
| Total costs of vaccination, A$ ('000) | 31,820.48 | 32,177.70 | 31,625.53 | 31,659.21 | 33,615.78 | 34,632.34 | 33,048.30 | 33,168.74 |
| Total treatment costs averted, A$ ('000) | 3,709.75 | 3,709.75 | 3,709.75 | 3,709.75 | 6,898.41 | 6,898.41 | 6,898.41 | 6,898.41 |
| Net costs of the vaccination, A$ ('000) | 28,110.73 | 28,467.95 | 27,915.78 | 27,949.47 | 26,717.37 | 27,733.93 | 26,149.90 | 26,270.33 |
| Number of averted- | | | | | | | | |
| - Cervical cancers case averted | 113 | 113 | 113 | 113 | 113 | 113 | 113 | 113 |
| - Deaths averted | 23 | 23 | 23 | 23 | 23 | 23 | 23 | 23 |
| - Life years saved | 543 | 543 | 543 | 543 | 543 | 543 | 543 | 543 |
| Nonfatal *DALYs* averted | 55 | 55 | 55 | 55 | 55 | 55 | 55 | 55 |
| Incremental cost-effectiveness ratio (*ICER*) per- | | | | | | | | |
| - Cervical cancers case averted, A$ | 248,767 | 251,929 | 247,042 | 247,340 | 236,437 | 245,433 | 231,415 | 232,481 |
| - Life saved, A$ | 1,222,205 | 1,237,737 | 1,213,730 | 1,215,194 | 1,161,625 | 1,205,823 | 1,136,952 | 1,142,188 |
| - Life year saved[1], A$ | 51,769 | 52,427 | 51,410 | 51,472 | 49,203 | 51,075 | 48,158 | 48,380 |
| - *DALYs* averted[1], A$ | 47,008 | 47,605 | 46,682 | 46,738 | 44,678 | 46,378 | 43,729 | 43,930 |
| Cost-effectiveness threshold | | | | | | | | |
| *GDP* per capita, A$ | 73,267 | 73,267 | 73,267 | 73,267 | 73,267 | 73,267 | 73,267 | 73,267 |
| Decision rules | | | | | | | | |
| - Very cost-effective[1] | Yes | Yes | Yes | Yes | Yes | Yes | Yes | Yes |
| - Cost-effective[2] | | | | | | | | |
| - No cost-effective[3] | | | | | | | | |

[1]Very cost-effective if *ICER* per *DALYs* averted < 1 time *GDP* per capita

[2]cost-effective if *ICER* per *DALYs* averted ≥ 1 times *GDP* per capita and ≤ 3 times *GDP* per capita

[3]no cost-effective if *ICER* per *DALYs* averted > 3 times *GDP* per capita.

\*Costs and *DALYs* were discounted at 5% per year.

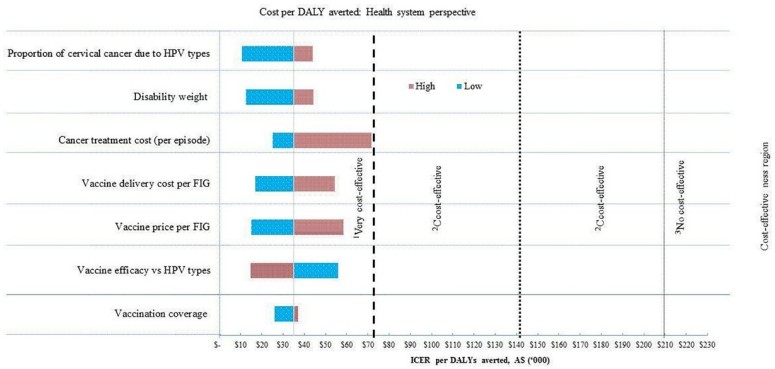

**Fig 2. Changes in input model parameters on ICER per DALY averted from a health system perspective.**

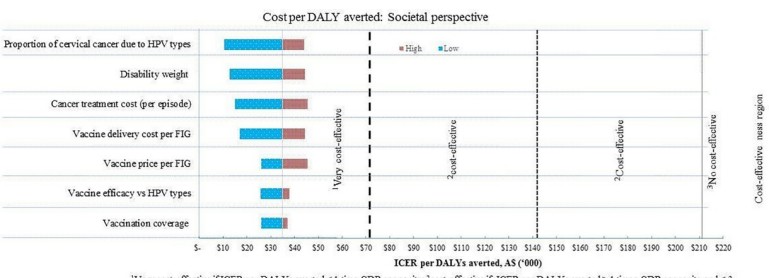

**Fig 3. Changes in input model parameters on ICER per DALY averted from a societal perspective.**

## Sensitivity analysis

Model uncertainty was investigated by changing the values of input parameters in the cost-effectiveness model from the health system (Fig 2) and societal perspectives (Fig 3). The output of the deterministic sensitivity analysis showed that the price of vaccine, vaccine delivery costs, the incidence of cervical cancer, vaccination coverage, vaccine efficacy, and cervical cancer treatment costs were the dominating parameters that influence the *ICERs* per *DALY* averted. These findings are conservative, as only a simple static model of 9vHPV vaccination was considered. According to the *WHO-CHOICE* threshold, the *9vHPV* vaccination is a very cost-effective and favourable option for introduction in Australia. This analysis indicates that the model outputs are robust to variation in the values of all parameters; however, there is a necessity to confirm that the pricing of the vaccine is appropriate in the context of Australia.

## Discussion

The present study is a comprehensive cost-effectiveness evaluation of the introduction of a *NIP* with the new *9vHPV* vaccine to adolescent girls in Australia. The impact of *9vHPV* vaccination on health and economic outcomes was measured using various model scenarios allowing for the testing of three different vaccine delivery strategies.

The findings show that the new 9-valent vaccination of 12-year-old adolescent girls is highly cost-effective, with *ICERs* per *DALY* of A$47,008 and A$44,678, from the health system and societal perspectives, respectively. Although the *9vHPV* vaccination has been implemented as part of the school-based delivery strategy, the present study has emphasised two other hypothetical delivery strategies, namely health facility-based and outreach-based programs. If the *9vHPV* vaccination program is extended to these delivery outlets, the *ICER* remains highly cost-effective at A$46,682/*DALY* averted for health facility-based and A$46,738/*DALY* averted for outreach-based vaccination programs compared with a school-based vaccination program (*ICER* = A$47,605/*DALY* averted). Considering the societal perspective, the *9vHPV* vaccination also reports a very cost-effective outcome, with an *ICER* of A$46,378/*DALY* averted, A$43,729/*DALY* averted, and A$43,930/*DALY* averted for the school-based, health facility-based and outreach-based vaccination programs, respectively. It is noteworthy that the *ICERs* are comparatively lower from the societal perspective in terms of vaccine delivery strategies compared with the health system perspective.

Immunisation would still be very cost-effective from both the health system and societal perspectives if the program is extended to encompass other delivery strategies. However, no herd immunity was considered in the context of these strategies. This evaluation provides a piece of initial evidence for the value of money of investments in the *9vHPV* vaccination and protection against transient and persistent infections of *HPV*. Under the input model

assumptions, the present evaluation of the two-dose *9vHPV* vaccination would be very cost-effective across delivery strategies. From the societal perspective, the *ICER* per *DALYs* averted was comparatively lower than the health system perspective in terms of delivery strategies. The cost-effectiveness evaluation is significant even allowing for different vaccine delivery strategies and vaccination model assumptions.

This study findings are consistent with the conclusions from the evaluation of cost-effectiveness of the *9vHPV* vaccination in other country settings including Austria [62], Canada [19], Germany [28], Italy [5], Kenya and Uganda [30], South Africa [52], and the *US* [21,63]. These studies estimated that an immunisation programs with the *9vHPV* vaccine was likely to belong within an acceptance heuristic threshold level of cost-effectiveness or even reach cost-saving status in different country settings. In Canada, the *9vHPV* vaccine was offered to school-aged girls and evidenced to be cost-effective at a price increment lower than CAN$24 [19]. Further, *9vHPV* was found to be cost-effective in the *US*, if the incremental cost per dose of the *9vHPV* was less than US$13 for a gender-neutral strategy (school-aged girls only) from a health system perspective [64]. From a societal perspective, the *9vHPV* vaccine would also be considered very cost-effective at the national and state levels in the *US* if the vaccine price of *9vHPV* was US$148 per dose (in 2016) [65], whereas two-dose schedules of the *9vHPV* vaccine were likely more cost-efficient compared with three-dose schedules considering the population-level effectiveness [18]. Another recent study showed that introducing a universal *9vHPV* vaccination in Germany would yield noteworthy incremental public health benefits and be highly cost-effective [28].

The present evaluation was performed among school-aged preadolescent girls (i.e., 12 years of age). Previous studies confirmed that vaccination of girls only was commonly more effective versus vaccination of both genders in different settings [28,32]. The two-dose *9vHPV* vaccination approach is recommended for the target cohort of adolescent girls aged 12 to 14 years for several reasons [39]. Giving the vaccination at this age is likely to ensure immunization before their first sexual encounter and *HPV* exposure. As a result, the immune response tends to be stronger and more long-lasting when the vaccine is present in preadolescent girls. A vaccination schedule against *HPV* would allow for a more efficient primary strategy by protecting females exposed to male partners and unvaccinated females to prevent *HPV* transmission [9,28,63]. Eventually it would provide additional benefits to potentially accomplish virus eradication [28].

Most previous studies pay little attention to comparing the cost-effectiveness from the health system and societal point of views across vaccine delivery strategies. Thus, the evidence produced is not sufficient for health policymakers to decide upon effective or conclusive strategies. This study findings however provides effective and efficient empirical evidence of its economic viability. Health policymakers can use this evidence for the allocation of health resources and extend their vaccination program to other country settings to ensure optimal health gains.

This study has some strengths that should be highlighted. This vaccination is justified overall by epidemiological and health and economic outcomes. Under the input model assumptions, this study demonstrates that the *9vHPV* vaccination is economically viable from both the health system and societal perspectives. A broader societal perspective calculates additional benefits of the new vaccine that are mainly associated with reduced productivity losses. *HPV*-related cervical lesions lead to a loss or reduction of women's household income due to high productivity loss (presenteeism) and absenteeism [66]. Ultimately, *HPV* related diseases lead to a decrease in a victim's socioeconomic position, which is costly for working women, their employers, and the economy. The study findings show distinctly that three vaccine delivery strategies (e.g., school-based, health facilities and outreach-based) are cost-effective. This is

significant for health policymakers, strategic leaders, health scientists, cancer experts and public health professionals to help promote further implementation and extension of vaccination via a universal immunisation strategy.

This study also has some caveats. Little evidence is available on the health and economic burden of cervical cancer in Australia. Some of the model parameters related to indirect costs for cervical cancer treatment and costs of vaccination across vaccine delivery strategies (e.g., school-based, health facility-based, and outreach-based) are not available for Australia. Indirect costs of patients (e.g., opportunity costs) in terms of absenteeism due to cervical cancer and caregiver time were taken from the academic literature and anecdotal evidence in Australia and international sources. In this context, the cost-of-illness study would be appropriate for measuring the productivity losses of patients and their caregivers. However, due to a limited timeframe it was not able to conduct a cost-of-illness study among cervical cancer patients. It was presumed that the *9vHPV* vaccine would be delivered to both boys and girls, but that it would only be cost-effective among girls, as the direct health impacts for *9vHPV* is expected to be small for boys. This study used the *GDP* per capita thresholds level as defined by *CMH*. The *GDP* threshold might be a suitable screening method but should not be the only consideration for vaccination investment as there are other issues such as feasibility, affordability, alternative interventions and other local considerations which are not accounted for in the threshold level decision rule. Finally, the study findings were generated for the national context in Australia and might vary by state or regional settings, depending on cervical cancer outcomes (e.g., incidence, mortality), treatment procedures, cancer stages, costs of vaccination, and coverage of immunisation.

## Conclusions

This study is an extensive cost-effectiveness analysis of *9vHPV* vaccination in Australia from both the health system and societal perspectives. The introduction of the *9vHPV* immunisation is assessed to be very cost-effective from both perspectives. It incorporated three delivery strategies (school-based, health facility-based, and outreach-based). However, this high-value vaccination would need substantial upfront investments. Considering a two-dose schedule, the *9vHPV* vaccination demonstrated 'good value for money', if the vaccination could accomplish a high vaccination coverage and provide protection. The findings of this evaluation contribute to decision-making about the incorporation of the *9vHPV* vaccine into a universal cervical cancer vaccination program in Australia. With continued assessment of the potential vaccine properties as well as vaccine delivery and scale-up strategies, the two-dose *9vHPV* vaccine would provide significant health and economic benefits for preadolescents and society. Finally, the success of *9vHPV* vaccination will be contingent on several predominating factors including value for money, feasibility, acceptability, and affordability.

## Supporting information

**S1 Table. Cost analysis.**
(XLSX)

## Acknowledgments

The study is part of the first author's PhD research works. The PhD program was funded by the University of Southern Queensland, Australia. We would also like to thank the Australian Institute of Health and Welfare, Central Cancer Registry, Australian Bureau of Statistics, and Australian Burden of Disease Study. We would like to gratefully acknowledge the reviewers

and editors of our manuscript. Special thanks also to my PhD fellow colleague (Syed Afroz Keramat) for his cordial support during data collection and model selection.

## Author Contributions

**Conceptualization:** Rashidul Alam Mahumud.

**Data curation:** Rashidul Alam Mahumud.

**Formal analysis:** Rashidul Alam Mahumud.

**Investigation:** Rashidul Alam Mahumud, Jeff Dunn, Jeff Gow.

**Methodology:** Rashidul Alam Mahumud.

**Project administration:** Rashidul Alam Mahumud, Khorshed Alam, Jeff Dunn, Jeff Gow.

**Resources:** Rashidul Alam Mahumud, Khorshed Alam, Jeff Dunn, Jeff Gow.

**Software:** Rashidul Alam Mahumud.

**Supervision:** Khorshed Alam, Jeff Dunn, Jeff Gow.

**Validation:** Rashidul Alam Mahumud, Khorshed Alam, Jeff Gow.

**Visualization:** Rashidul Alam Mahumud, Khorshed Alam.

**Writing – original draft:** Rashidul Alam Mahumud, Khorshed Alam, Jeff Dunn, Jeff Gow.

**Writing – review & editing:** Rashidul Alam Mahumud, Khorshed Alam, Jeff Dunn, Jeff Gow.

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
