## [Decision Letter · Decision Letter 0]

29 Aug 2019

[EXSCINDED]

PONE-D-19-21887

The cost-effectiveness of controlling cervical cancer using a new 9-valent human papillomavirus vaccine among school-aged girls in Australia

PLOS ONE

Dear Mr. Mahumud,

Thank you for submitting your manuscript to PLOS ONE. After careful consideration, we feel that it has merit but does not fully meet PLOS ONE’s publication criteria as it currently stands. Therefore, we invite you to submit a revised version of the manuscript that addresses all points raised during the review process.

We would appreciate receiving your revised manuscript by Oct 13 2019 11:59PM. To enhance the reproducibility of your results, we recommend that if applicable you deposit your laboratory protocols in protocols.io, where a protocol can be assigned its own identifier (DOI) such that it can be cited independently in the future. For instructions see: http://journals.plos.org/plosone/s/submission-guidelines#loc-laboratory-protocols

We look forward to receiving your revised manuscript.

Kind regards,

Marcia Edilaine Lopes Consolaro, Ph.D

Academic Editor

PLOS ONE

Journal Requirements:

2. Please clearly list in your methods section or as supporting information,  all the data sources and corresponding references used in this study. Additionally, if you collected any data that were not obtained from previously published research, provide details as to where these data are from and how they were collected.

3. We ask that you please use a more technically clear term to replace "sexual debut" as this not well defined in the manuscript and could be misinterpreted.

Reviewers' comments:

Reviewer's Responses to Questions

**Comments to the Author**

1. Is the manuscript technically sound, and do the data support the conclusions?

Reviewer #1: Yes

Reviewer #2: Yes

2. Has the statistical analysis been performed appropriately and rigorously? 

Reviewer #1: N/A

Reviewer #2: Yes

3. Have the authors made all data underlying the findings in their manuscript fully available?

Reviewer #1: Yes

Reviewer #2: Yes

4. Is the manuscript presented in an intelligible fashion and written in standard English?

Reviewer #1: Yes

Reviewer #2: Yes

5. Review Comments to the Author

Reviewer #1: I found the work interesting and very detailed. Overall it was quite comprehensive, evaluating many parameters that may affect cost-effectiveness studies. My only suggestion concerns the quality of the figures, I believe they could improve a little more.

Reviewer #2: The objective of this study was to examine the cost-effectiveness of adding a 9vHPV vaccine to the national immunization schedule in Australia across three different delivery strategies. The study is very interesting and important. The methods and results are described in detail. The conclusions are presented in an appropriate fashion and are supported by the data. The article is presented in an intelligible fashion.

I would like make some observations:

1- In the objectives (page 5, line 26) the term “DALYs” is used by first time, please describe this acronym in this moment.

2- In the results (Page 12, line 11) the acronym “IECR” should be substituted by “ICER”?

6. PLOS authors have the option to publish the peer review history of their article (what does this mean?). If published, this will include your full peer review and any attached files.

Reviewer #1: No

Reviewer #2: No

---

## [Author Response · Author response to Decision Letter 0]

10 Sep 2019

Response to Reviewer #1 comments:

Comment: I found the work interesting and very detailed. Overall it was quite comprehensive, evaluating many parameters that may affect cost-effectiveness studies. My only suggestion concerns the quality of the figures, I believe they could improve a little more.

Response: Authors express gratitude to the reviewer for their appreciation. The figures have been revised as per journal requirements. Please see the revised figures.

Response to Reviewer #2 comments:

Comment-1: 

The objective of this study was to examine the cost-effectiveness of adding a 9vHPV vaccine to the national immunization schedule in Australia across three different delivery strategies. The study is very interesting and important. The methods and results are described in detail. The conclusions are presented in an appropriate fashion and are supported by the data. The article is presented in an intelligible fashion.

Response: Authors express gratitude to the reviewer for their appreciation.

Comment-2: I would like make some observations: In the objectives (page 5, line 26) the term “DALYs” is used by first time, please describe this acronym in this moment.

Response: The text has been revised now by describing the term DALY (i.e., disability adjusted-life years) as advised by the reviewer. Please see page 6 (lines 7-8).

Comment-3: In the results (Page 12, line 11) the acronym “IECR” should be substituted by “ICER”?

 Response: Authors are thankful to the reviewer for raising the issue. We have corrected the acronym (i.e., ICER). Please see page 12 (line-26).

---

## [Decision Letter · Decision Letter 1]

23 Sep 2019

PONE-D-19-21887R1

The cost-effectiveness of controlling cervical cancer using a new 9-valent human papillomavirus vaccine among school-aged girls in Australia

PLOS ONE

Dear Mr. Mahumud,

Thank you for submitting your manuscript to PLOS ONE. After careful consideration, we feel that it has merit but does not fully meet PLOS ONE’s publication criteria as it currently stands. Therefore, we invite you to submit a revised version of the manuscript that addresses all points raised during the review process.

We would appreciate receiving your revised manuscript by Nov 07 2019 11:59PM. To enhance the reproducibility of your results, we recommend that if applicable you deposit your laboratory protocols in protocols.io, where a protocol can be assigned its own identifier (DOI) such that it can be cited independently in the future. For instructions see: http://journals.plos.org/plosone/s/submission-guidelines#loc-laboratory-protocols

We look forward to receiving your revised manuscript.

Kind regards,

Marcia Edilaine Lopes Consolaro, Ph.D

Academic Editor

PLOS ONE

Reviewers' comments:

Reviewer's Responses to Questions

**Comments to the Author**

1. If the authors have adequately addressed your comments raised in a previous round of review and you feel that this manuscript is now acceptable for publication, you may indicate that here to bypass the “Comments to the Author” section, enter your conflict of interest statement in the “Confidential to Editor” section, and submit your "Accept" recommendation.

Reviewer #1: All comments have been addressed

Reviewer #2: All comments have been addressed

2. Is the manuscript technically sound, and do the data support the conclusions?

Reviewer #1: Yes

Reviewer #2: Yes

3. Has the statistical analysis been performed appropriately and rigorously? 

Reviewer #1: Yes

Reviewer #2: I Don't Know

4. Have the authors made all data underlying the findings in their manuscript fully available?

Reviewer #1: Yes

Reviewer #2: Yes

5. Is the manuscript presented in an intelligible fashion and written in standard English?

Reviewer #1: Yes

Reviewer #2: Yes

6. Review Comments to the Author

Reviewer #1: After reviewing the work again, I noticed small issues that can be easily fixed:

1- On p. 2, line 7 and p. 3, line 23, please replace genotype by type, it is the most correct term today to refer to HPV types.

2- On p. 5, line 12, replace the term nonavalent-HPV vaccine with the abbreviation (9vHPV) that has been previously used.

3 - On p. 7, line 30, describe the term PBAC.

4 - On p. 8, line 6 describe the term WTP.

5 - On p. 8, line 23 describe the term NIP.

6 - On p. 16, line 20, separate the words "costper".

7 - On p. 18, line 5, separate the words "effectiveamong".

Reviewer #2: (No Response)

7. PLOS authors have the option to publish the peer review history of their article (what does this mean?). If published, this will include your full peer review and any attached files.

Reviewer #1: No

Reviewer #2: No

---

## [Author Response · Author response to Decision Letter 1]

23 Sep 2019

Response to the editor’s and reviewer’s comments

Date: 23 September 2019 

Dear Reviewer,

Thank you for giving us an opportunity to revise our manuscript entitled “The cost-effectiveness of controlling cervical cancer using a new 9-valent human papillomavirus vaccine among school-aged girls in Australia”. We found the reviewers’ comments/feedback very helpful in improving the manuscript and we have revised the manuscript accordingly. Please find attached the revised manuscript. We declare that all authors have no conflicts of interest. The manuscript has not been published in any other journal. Our point-by-point comments on the suggested revisions are below. 

Best regards,

Rashidul Alam Mahumud (corresponding author)

PhD Candidate, MPH, MSc

On behalf of all of the co-authors

Health Economics Research, 

Health Systems and Population Studies Division, 

International Centre for Diarrhoeal Disease Research, Bangladesh (icddr,b), 

Dhaka-1212, Bangladesh.

Response to Reviewer #1 comments:

Comment 1. All comments have been addressed. After reviewing the work again, I noticed small issues that can be easily fixed:

Response: We found the reviewers’ comments/feedback very helpful in improving the manuscript and we have revised the manuscript accordingly (please see the revised manuscript).

Comment 2. On p. 2, line 7 and p. 3, line 23, please replace genotype by type, it is the most correct term today to refer to HPV types.

Response: Thank you for your valuable concerns. The text has been corrected now accordingly (please see on page 2 (line-7) and page 3 (line-23)).

Comment 3. On p. 5, line 12, replace the term nonavalent-HPV vaccine with the abbreviation (9vHPV) that has been previously used.

Response: The term “nonavalent-HPV vaccine” has been replaced by “9vHPV” (please see on page 5, line-12).

Comment 4. On p. 7, line 30, describe the term PBAC.

Response: The text has been revised by describing the term PBAC as defined “Pharmaceutical Benefits Advisory Committee” (please see on page 7, lines-28-29).

Comment 5. On p. 8, line 6 describe the term WTP.

Response: The term WTP has been explained as “Willingness-to-pay” (please see page on 8, lines-4-5).

Comment 6. On p. 8, line 23 describe the term NIP.

Response: The term NIP has been described as “national immunisation program” (please see on page 8, line-21).

Comment 7. On p. 16, line 20, separate the words "costper".

Response: Corrected. Please see on page 16, line-19-20.

Comment 8. On p. 18, line 5, separate the words "effectiveamong".

Response: Corrected. Please see on page 18 (line 5).

Response to Reviewer #2 comments:

Comment-1: All comments have been addressed

Response: Authors express gratitude to the reviewer for their appreciation.

---

## [Decision Letter · Decision Letter 2]

26 Sep 2019

The cost-effectiveness of controlling cervical cancer using a new 9-valent human papillomavirus vaccine among school-aged girls in Australia

PONE-D-19-21887R2

Dear Dr. Mahumud,

We are pleased to inform you that your manuscript has been judged scientifically suitable for publication and will be formally accepted for publication once it complies with all outstanding technical requirements.

With kind regards,

Marcia Edilaine Lopes Consolaro, Ph.D

Academic Editor

PLOS ONE

Reviewers' comments:

Reviewer's Responses to Questions

**Comments to the Author**

1. If the authors have adequately addressed your comments raised in a previous round of review and you feel that this manuscript is now acceptable for publication, you may indicate that here to bypass the “Comments to the Author” section, enter your conflict of interest statement in the “Confidential to Editor” section, and submit your "Accept" recommendation.

Reviewer #1: All comments have been addressed

2. Is the manuscript technically sound, and do the data support the conclusions?

Reviewer #1: Yes

3. Has the statistical analysis been performed appropriately and rigorously? 

Reviewer #1: Yes

4. Have the authors made all data underlying the findings in their manuscript fully available?

Reviewer #1: Yes

5. Is the manuscript presented in an intelligible fashion and written in standard English?

Reviewer #1: Yes

6. Review Comments to the Author

Reviewer #1: (No Response)

7. PLOS authors have the option to publish the peer review history of their article (what does this mean?). If published, this will include your full peer review and any attached files.

Reviewer #1: No

---

## [Editor Report · Acceptance letter]

1 Oct 2019

PONE-D-19-21887R2 

The cost-effectiveness of controlling cervical cancer using a new 9-valent human papillomavirus vaccine among school-aged girls in Australia 

Dear Dr. Mahumud:

I am pleased to inform you that your manuscript has been deemed suitable for publication in PLOS ONE. Congratulations! Your manuscript is now with our production department. 

With kind regards,

on behalf of

Professor Marcia Edilaine Lopes Consolaro 

Academic Editor

PLOS ONE